# MODEL COMPRESSION VIA DISTILLATION AND QUANTIZATION

**Antonio Polino**
ETH Zürich
antonio.polino1@gmail.com

**Razvan Pascanu**
Google DeepMind
razp@google.com

**Dan Alistarh**
IST Austria
dan.alistarh@ist.ac.at

## ABSTRACT

Deep neural networks (DNNs) continue to make significant advances, solving tasks from image classification to translation or reinforcement learning. One aspect of the field receiving considerable attention is efficiently executing deep models in resource-constrained environments, such as mobile or embedded devices. This paper focuses on this problem, and proposes two new compression methods, which jointly leverage *weight quantization* and *distillation* of larger networks, called "teachers," into compressed "student" networks. The first method we propose is called *quantized distillation* and leverages distillation during the training process, by incorporating distillation loss, expressed with respect to the teacher network, into the training of a smaller student network whose weights are quantized to a limited set of levels. The second method, *differentiable quantization*, optimizes the *location of quantization points* through stochastic gradient descent, to better fit the behavior of the teacher model. We validate both methods through experiments on convolutional and recurrent architectures. We show that quantized shallow students can reach similar accuracy levels to state-of-the-art full-precision teacher models, while providing up to order of magnitude compression, and inference speedup that is almost linear in the depth reduction. In sum, our results enable DNNs for resource-constrained environments to leverage architecture and accuracy advances developed on more powerful devices.

## 1 INTRODUCTION

**Background.** Neural networks are extremely effective for solving several real world problems, like image classification (Krizhevsky et al., 2012; He et al., 2016a), translation (Vaswani et al., 2017), voice synthesis (Oord et al., 2016) or reinforcement learning (Mnih et al., 2013; Silver et al., 2016). At the same time, modern neural network architectures are often compute, space and power hungry, typically requiring powerful GPUs to train and evaluate. The debate is still ongoing on whether large models are *necessary* for good accuracy. It is known that individual network weights can be redundant, and may not carry significant information, e.g. Han et al. (2015). At the same time, large models often have the ability to completely memorize datasets (Zhang et al., 2016), yet they do not, but instead appear to learn generic task solutions. A standing hypothesis for why overcomplete representations are necessary is that they make learning possible by transforming local minima into saddle points (Dauphin et al., 2014) or to discover *robust* solutions, which do not rely on precise weight values (Hochreiter & Schmidhuber, 1997; Keskar et al., 2016).

If large models are only needed for robustness during training, then significant compression of these models should be achievable, without impacting accuracy. This intuition is strengthened by two related, but slightly different research directions. The first direction is the work on *training quantized neural networks*, e.g. Courbariaux et al. (2015); Rastegari et al. (2016); Hubara et al. (2016); Wu et al. (2016a); Mellempudi et al. (2017); Ott et al. (2016); Zhu et al. (2016), which showed that neural networks can converge to good task solutions even when weights are constrained to having values from a set of integer levels. The second direction aims to *compress* already-trained models, while preserving their accuracy. To this end, various elegant compression techniques have been

proposed, e.g. Han et al. (2015); Iandola et al. (2016); Wen et al. (2016); Gysel et al. (2016); Mishra et al. (2017), which combine quantization, weight sharing, and careful coding of network weights, to reduce the size of state-of-the-art deep models by orders of magnitude, while at the same time speeding up inference.

Both these research directions are extremely active, and have been shown to yield significant compression and accuracy improvements, which can be crucial when making such models available on embedded devices or phones. However, the literature on compressing deep networks focuses almost exclusively on finding good compression schemes for a given model, without significantly altering the *structure of the model*. On the other hand, recent parallel work (Ba & Caruana, 2013; Hinton et al., 2015) introduces the process of *distillation*, which can be used for transferring the behaviour of a given model to any other structure. This can be used for compression, e.g. to obtain compact representations of ensembles (Hinton et al., 2015). However the size of the student model needs to be large enough for allowing learning to succeed. A model that is too shallow, too narrow, or which misses necessary units, can result in considerable loss of accuracy (Urban et al., 2016).

In this work, we examine whether distillation and quantization can be jointly leveraged for better compression. We start from the intuition that 1) the existence of highly-accurate, full-precision teacher models should be leveraged to improve the performance of quantized models, while 2) quantizing a model can provide better compression than a distillation process attempting the same space gains by purely decreasing the number of layers or layer width. While our approach is very natural, interesting research questions arise when these two ideas are combined.

**Contribution.** We present two methods which allow the user to compound compression in terms of *depth*, by distilling a *shallower student* network with similar accuracy to a *deeper teacher* network, with compression in terms of *width*, by quantizing the weights of the student to a limited set of integer levels, and using less weights per layer. The basic idea is that quantized models can leverage *distillation loss* (Hinton et al., 2015), the weighted average between the correct targets (represented by the labels) and soft targets (represented by the teacher's outputs).

We implement this intuition via two different methods. The first, called *quantized distillation*, aims to leverage distillation loss during the training process, by incorporating it into the training of a student network whose weights are constrained to a limited set of levels. The second method, which we call *differentiable quantization*, takes a different approach, by attempting to converge to the *optimal location of quantization points* through stochastic gradient descent. We validate both methods empirically through a range of experiments on convolutional and recurrent network architectures. We show that quantized shallow students can reach similar accuracy levels to full-precision and deeper teacher models on datasets such as CIFAR and ImageNet (for image classification) and OpenNMT and WMT (for machine translation), while providing up to order of magnitude compression, and inference speedup that is linear in the depth. [1]

**Related Work.** Our work is a special case of *knowledge distillation* (Ba & Caruana, 2013; Hinton et al., 2015), in which we focus on techniques to obtain high-accuracy students that are both quantized and shallower. More generally, it can be seen as a special instance of *learning with privileged information*, e.g. Vapnik & Izmailov (2015); Xu et al. (2016), in which the student is provided additional information in the form of outputs from a larger, pre-trained model. The idea of optimizing the locations of quantization points during the learning process, which we use in differentiable quantization, has been used previously in Lan et al. (2014); Koren & Sill (2011); Zhang et al. (2017), although in the different context of matrix completion and recommender systems.

Using distillation for size reduction is mentioned in Hinton et al. (2015), for distilling ensembles. To our knowledge, the only other work using distillation in the context of quantization is Wu et al. (2016b), which uses it to improve the accuracy of binary neural networks on ImageNet. We significantly refine this idea, as we match or even improve the accuracy of the original full-precision model: for example, our 4-bit quantized version of ResNet18 has *higher accuracy* than full-precision ResNet18 (matching the accuracy of the ResNet34 teacher): it has higher top-1 accuracy (by >15%) and top-5 accuracy (by >7%) compared to the most accurate model in Wu et al. (2016b).

---

[1]Source code available at `https://github.com/antspy/quantized_distillation`

## 2 PRELIMINARIES

### 2.1 THE QUANTIZATION PROCESS

We start by defining a *scaling function sc* $: \mathcal{R}^n \to [0, 1]$, which normalizes vectors whose values come from an arbitrary range, to vectors whose values are in $[0, 1]$. Given such a function, the general structure of the quantization functions is as follows:

$$Q(v) = sc^{-1} \left( \hat{Q} \left( sc(v) \right) \right), \tag{1}$$

where $sc^{-1}$ is the inverse of the scaling function, and $\hat{Q}$ is the actual quantization function that only accepts values in $[0, 1]$. We always assume $v$ to be a vector; in practice, of course, the weight vectors can be multi-dimensional, but we can reshape them to one dimensional vectors and restore the original dimensions after the quantization.

**Scaling.** There are various specifications for the scaling function; in this paper, we will use *linear scaling*, e.g. He et al. (2016b), that is $sc(v) = \frac{v - \beta}{\alpha}$, with $\alpha = \max_i v_i - \min_i v_i$ and $\beta = \min_i v_i$ which results in the target values being in $[0, 1]$, and the quantization function

$$Q(v) = \alpha \hat{Q} \left( \frac{v - \beta}{\alpha} \right) + \beta. \tag{2}$$

**Bucketing.** One problem with this formulation is that an identical scaling factor is used for the whole vector, whose dimension might be huge. Magnitude imbalance can result in a significant loss of precision, where most of the elements of the scaled vector are pushed to zero. To avoid this, we will use *bucketing*, e.g. Alistarh et al. (2016), that is, we will apply the scaling function separately to buckets of consecutive values of a certain fixed size. The trade-off here is that we obtain better quantization accuracy for each bucket, but will have to store two floating-point scaling factors for each bucket. We characterize the compression comparison in Section 5. The function $\hat{Q}$ can also be defined in several ways. We will consider both uniform and non-uniform placement of quantization points.

**Uniform Quantization.** We fix a parameter $s \geq 1$, describing the number of quantization levels employed. Intuitively, uniform quantization considers $s + 1$ equally spaced points between 0 and 1 (including these endpoints). The deterministic version will assign each (scaled) vector coordinate $v_i$ to the closest quantization point, while in the stochastic version we perform rounding probabilistically, such that the resulting value is an unbiased estimator of $v_i$, of minimal variance.

Formally, the uniform quantization function with $s + 1$ levels is defined as

$$\hat{Q}(v, s)_i = \frac{\lfloor v_i s \rfloor}{s} + \frac{\xi_i}{s}, \tag{3}$$

where $\xi_i$ is the rounding function. For the deterministic version, we define $k_i = sv_i - \lfloor v_i s \rfloor$ and set

$$\xi_i = \begin{cases} 1, & \text{if } k_i > \frac{1}{2} \\ 0, & \text{otherwise,} \end{cases} \tag{4}$$

while for the stochastic version we will set $\xi_i \sim Bernoulli(k_i)$. Note that $k_i$ is the normalized distance between the original point $v_i$ and the closest quantization point that is smaller than $v_i$ and that the vector components are quantized independently.

**Non-Uniform Quantization.** Non-uniform quantization takes as input a set of $s$ quantization points $\{p_1, \ldots, p_s\}$ and quantizes each element $v_i$ to the closest of these points. For simplicity, we only define the *deterministic* version of this function.

### 2.2 STOCHASTIC QUANTIZATION IS EQUIVALENT TO ADDING GAUSSIAN NOISE

In this section we list some interesting mathematical properties of the uniform quantization function. Clearly, stochastic uniform quantization is an unbiased estimator of its input, i.e. $E[Q(v)] = v$.

What interests us is applying this function to neural networks; as the scalar product is the most common operation performed by neural networks, we would like to study the properties of $Q(v)^T x$, where $v$ is the weight vector of a certain layer in the network and $x$ are the inputs. We are able to show that

$$Q(v)^T x = v^T x + \varepsilon \qquad (5)$$

where $\varepsilon$ is a random variable that is asymptotically normally distributed, i.e. $\frac{1}{\sigma_n}\varepsilon \xrightarrow{\mathcal{D}} \mathcal{N}(0, 1)$. Convergence occurs with the dimension $n$. For a formal statement and proof, see Section B.1 in the Appendix.

This means that quantizing the weights is equivalent to adding to the output of each layer (before the activation function) a zero-mean error term that is asymptotically normally distributed. The variance of this error term depends on $s$. This connects quantization to work advocating adding noise to intermediary activations of neural networks as a regularizer (Gulcehre et al., 2016) and to Arora et al. (2018), which investigates the connection between adding noise to a network weights and the network generalization properties. We plan to investigate this connection in more detail in future work.

## 3 QUANTIZED DISTILLATION

The context is the following: given a task, we consider a trained state-of-the-art deep model solving it–the *teacher*, and a compressed *student* model. The student is compressed in the sense that 1) it is *shallower* than the teacher; and 2) it is *quantized*, in the sense that its weights are expressed at limited bit width. The strategy, as for standard distillation (Ba & Caruana, 2013; Hinton et al., 2015) is for the student to leverage the converged teacher model to reach similar accuracy. We note that distillation has been used previously to obtain compact high-accuracy encodings of ensembles (Hinton et al., 2015); however, we believe this is the first time it is used for model compression via quantization.

Given this setup, there are two questions we need to address. The first is how to transfer knowledge from the teacher to the student. For this, the student will use the *distillation loss*, as defined by Hinton et al. (2015), as the weighted average between two objective functions: cross entropy *with soft targets*, controlled by the temperature parameter $T$, and the cross entropy *with the correct labels*. We refer the reader to Hinton et al. (2015) for the precise definition of distillation loss.

The second question is how to employ distillation loss in the context of a *quantized* neural network. An intuitive approach is to rely on projected gradient descent, where a gradient step is taken as in full-precision training, and then the new parameters are *projected* to the set of valid solutions. Critically, we accumulate the error at each projection step into the gradient for the next step. One can think of this process as if *collecting evidence for whether each weight needs to move to the next quantization point or not*. Crucially, the error accumulation prevents the algorithm from getting stuck in the current solution if gradients are small, which would occur in a naive projected gradient approach. This is similar to the approach taken by BinaryConnect technique, with some differences. Li et al. (2017) also examines these dynamics in detail. Compared to BinnaryConnect, we use distillation rather than learning from scratch, hence learning more efficiently. We also do not restrict ourselves to binary representation, but rather use *variable bit-width* quantization functions and *bucketing*, as defined in Section 2.

An alternative view of this process, illustrated in Figure 1, is that we perform the SGD step on the *full-precision* model, but computing the gradient on the *quantized model*, expressed with respect to the *distillation loss*. See Algorithm 1 for details.

## 4 DIFFERENTIABLE QUANTIZATION

### 4.1 GENERAL DESCRIPTION

We introduce differentiable quantization as a general method of improving the accuracy of a quantized neural network, by exploiting non-uniform quantization point placement. In particular, we are going to use the non-uniform quantization function defined in Section 2.1. Experimentally, we have

---

**Algorithm 1** Quantized Distillation

1: **procedure** QUANTIZED DISTILLATION
2:     *Let $w$ be the network weights*
3: *loop*
4:     $w^q \leftarrow$ quant_function$(w, s)$
5:     *Run forward pass and compute distillation loss $l(w^q)$*
6:     *Run backward pass and compute* $\frac{\partial l(w^q)}{\partial w^q}$
7:     *Update original weights using SGD **in full precision*** $w = w - \nu \cdot \frac{\partial l(w^q)}{\partial w^q}$
8: *Finally quantize the weights before returning:* $w^q \leftarrow$ quant_function$(w, s)$
9: **return** $w^q$

---

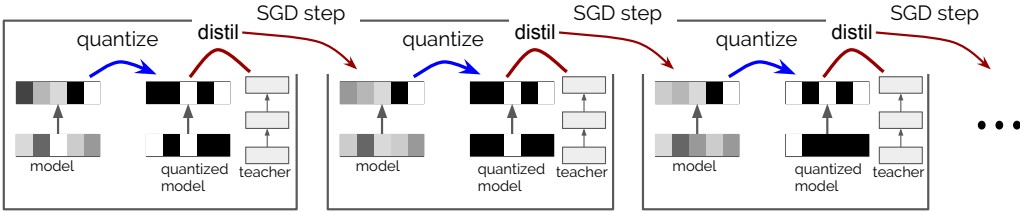

Figure 1: Depiction of the steps of quantized distillation. Note the accumulation over multiple steps of gradients in the unquantized model leads to a switch in quantization (e.g. top layer left most square).

found little difference between stochastic and deterministic quantization in this case, and therefore will focus on the simpler deterministic quantization function here.

Let $p = (p_1, \ldots, p_s)$ be the vector of quantization points, and let $Q(v, p)$ be our quantization function, as defined previously. Ideally, we would like to find a set of quantization points $p$ which minimizes the accuracy loss when quantizing the model using $Q(v, p)$. The key observation is that to find this set $p$, we can just use stochastic gradient descent, because we are able to compute the gradient of $Q$ with respect to $p$.

A major problem in quantizing neural networks is the fact that the decision of which $p_i$ should replace a given weight is discrete, hence the gradient is zero: $\dfrac{\partial Q(v, p)}{\partial v} = 0$, almost everywhere. This implies that we cannot backpropagate the gradients through the quantization function. To solve this problem, typically a variant of the straight-through estimator is used, see e.g. Bengio et al. (2013); Hubara et al. (2016). On the other hand, the model as a function of the chosen $p_i$ is *continuous* and *can be differentiated*; the gradient of $Q(v, p)_i$ with respect to $p_j$ is well defined almost everywhere, and it is simply

$$\frac{\partial Q(v, p)_i}{\partial p_j} = \begin{cases} \alpha_i, & \text{if } v_i \text{ has been quantized to } p_j \\ 0, & \text{otherwise,} \end{cases} \tag{6}$$

where $\alpha_i$ is $i$-th element of the scaling factor, assuming we are using a bucketing scheme. If no bucketing is used, then $\alpha_i = \alpha$ for every $i$. Otherwise it changes depending on which bucket the weight $v_i$ belongs to.

Therefore, we can use the same loss function we used when training the original model, and with Equation (6) and the usual backpropagation algorithm we are able to compute its gradient with respect to the quantization points $p$. Then we can minimize the loss function with respect to $p$ with the standard SGD algorithm. See Algorithm 2 for details.

**Note on Efficiency.** Optimizing the points $p$ can be slower than training the original network, since we have to perform the normal forward and backward pass, and in addition we need to quantize the weights of the model and perform the backward pass to get to the gradients w.r.t. $p$. However, in our experience differential quantization requires an order of magnitude less iterations to converge to a good solution, and can be implemented efficiently.

---

**Algorithm 2** Differentiable Quantization

1: **procedure** DIFFERENTIABLE QUANTIZATION
2:     *Let $w$ be the networks weights and $p$ the initial quantization points*
3: *loop*
4:     $w^q \leftarrow$ quant_function$(w, p)$
5:     *Run forward pass and compute loss $l(w^q)$*
6:     *Run backward pass and compute $\frac{\partial l(w^q)}{\partial w^q}$*
7:     *Use equation 6 to compute $\frac{\partial l(w^q)}{\partial p}$*
8:     *Update quantization points using SGD or similar: $p = p - \nu \cdot \frac{\partial l(w^q)}{\partial p}$*
9: **return** p

---

**Weight Sharing.** Upon close inspection, this method can be related to weight sharing Han et al. (2015). Weight sharing uses a $k$-mean clustering algorithm to find good clusters for the weights, adopting the centroids as quantization points for a cluster. The network is trained modifying the values of the centroids, aggregating the gradient in a similar fashion. The difference is in the initial assignment of points to centroids, but also, more importantly, in the fact that the assignment of weights to centroids never changes. By contrast, at every iteration we re-assign weights to the closest quantization point, and use a different initialization.

## 4.2 DISCUSSION AND ADDITIONAL HEURISTICS

While the loss is continuous w.r.t. $p$, there are indirect effects when changing the way each weight gets quantized. This can have drastic effect on the learning process. As an extreme example, we could have degeneracies, where all weights get represented by the same quantization point, making learning impossible. Or diversity of $p_i$ gets reduced, resulting in very few weights being represented at a really high precision while the rest are forced to be represented in a much lower resolution.

To avoid such issues, we rely on the following set of heuristics. Future work will look at adding a reinforcement learning loss for how the $p_i$ are assigned to weights.

**Choose good starting points.** One way to initialize the starting quantization points is to make them uniformly spaced, which would correspond to use as a starting point the uniform quantization function. The differentiable quantization algorithm needs to be able to use a quantization point in order to update it; therefore, to make sure every quantization point is used we initialize the points to be the quantiles of the weight values. This ensures that every quantization point is associated with the same number of values and we are able to update it.

**Redistribute bits where it matters.** Not all layers in the network need the same accuracy. A measure of how important each weight is to the final prediction is the norm of the gradient of each weight vector. So in an initial phase we run the forward and backward pass a certain number of times to estimate the gradient of the weight vectors in each layer, we compute the average gradient across multiple minibatches and compute the norm; we then allocate the number of points associated with each weight according to a simple linear proportion. In short we estimate

$$\left\| \mathbb{E}\left[\frac{\partial l}{\partial v}\right] \right\|_2 \tag{7}$$

where $l$ is the loss function, $v$ is the vector of weights in a particular layer and $\left(\frac{\partial l}{\partial v}\right)_i = \frac{\partial l}{\partial v_i}$ and we use this value to determine which layers are most sensitive to quantization.

When using this process, we will use more than the indicated number of bits in some layers, and less in others. We can reduce the impact of this effect with the use of Huffman encoding, see Section 5; in any case, note that while the total number of points stays constant, allocating more points to a layer will increase bit complexity overall if the layer has a larger proportion of the weights.

**Use the distillation loss.** In the algorithm delineated above, the loss refers to the loss we used to train the original model with. Another possible specification is to treat the unquantized model as the teacher model, the quantized model as the student, and to use as loss the distillation loss between the outputs of the unquantized and quantized model. In this case, then, we are optimizing our quantized model not to perform best with respect to the original loss, but to mimic the results of the unquantized model, which should be easier to learn for the model and provide better results.

**Hyperparameter optimization.** The algorithm above is an optimization problem very similar to the original one. As usual, to obtain the best results one should experiment with hyperparameters optimization, and different variants of gradient descent.

## 5 COMPRESSION

We now analyze the space savings when using $b$ bits and bucket size of $k$. Let $f$ be the size of full precision weights (32 bit) and let $N$ be the size of the "vector" we are quantizing. Full precision requires $fN$ bits, while the quantized vector requires $bN + \frac{2fN}{k}$. (We use $b$ bits per weight, plus the scaling factors $\alpha$ and $\beta$ for every bucket). The size gain is therefore $g(b, k; f) = \frac{kf}{kb+2f}$.

For differentiable quantization, we also have to store the values of the quantization points. Since this number does not depend on $N$, the amount of space required is negligible and we ignore it for simplicity. As an example, at 256 bucket size, using 2 bits per component yields $14.2\times$ space savings w.r.t. full precision, while 4 bits yields $7.52\times$ space savings. At 512 bucket size, the 2 bit savings are $15.05\times$, while 4 bits yields $7.75\times$ compression.

**Huffman encoding.** To save additional space, we can use Huffman encoding to represent the quantized values. In fact, each quantized value can be thought as the pointer to a full precision value; in the case of non uniform quantization is $p_k$, in the case of uniform quantization is $k/s$. We can then compute the frequency for every index across all the weights of the model and compute the optimal Huffman encoding. The mean bit length of the optimal encoding is the amount of bits we actually use to encode the values. This explains the presence of fractional bits in some of our size gain tables from the Appendix.

We emphasize that we only use these compression numbers as a ballpark figure, since additional implementation costs might mean that these savings are not always easy to translate to practice (Han et al., 2015).

## 6 EXPERIMENTAL RESULTS

### 6.1 SMALL DATASETS

**Methods.** We will begin with a set of experiments on smaller datasets, which allow us to more carefully cover the parameter space. We compare the performance of the methods described in the following way: we consider as baseline the *teacher* model, the *distilled* model and a *smaller* model: the distilled and smaller models have the same architecture, but the distilled model is trained using distillation loss on the teacher, while the smaller model is trained directly on targets. Further, we compare the performance of Quantized Distillation and Differentiable Quantization. In addition, we will also use PM ("post-mortem") quantization, which uniformly quantizes the weights after training without any additional operation, with and without bucketing. All the results are obtained with a bucket size of 256, which we found to empirically provide a good compression-accuracy trade-off. We refer the reader to Appendix A for details of the datasets and models.

**CIFAR-10 Experiments.** For image classification on CIFAR-10, we tested the impact of different training techniques on the accuracy of the distilled model, while varying the parameters of a CNN architecture, such as quantization levels and model size. Table 1 contains the results for full-precision training, PM quantization with and without bucketing, as well as our methods. The percentages on the left below the student models definition are the accuracy of the normal and the distilled model respectively (trained with full precision). More details are reported in table 11 in the appendix. We also tried an additional model where the student is deeper than the teacher, where we obtained that the student quantized to 4 bits is able to achieve significantly better accuracy than the teacher, with a compression factor of more than $7\times$.

We performed additional experiments for differentiable quantization using a wide residual network (Zagoruyko & Komodakis, 2016) that gets to higher accuracies; see table 3.

Overall, quantized distillation appears to be the method with best accuracy across the whole range of bit widths and architectures. It outperforms PM significantly for 2bit and 4bit quantization, achieves accuracy within $0.2\%$ of the *teacher* at 8 bits on the larger student model, and relatively minor accuracy loss at 4bit quantization. Differentiable quantization is a close second on all experiments,

Table 1: CIFAR10 accuracy. Teacher model: 5.3M param, 21 MB, accuracy 89.71 %.Details about the resulting size of the models are reported in table 11 in the appendix.

|  |  | 2 bits | 4 bits | 8 bits |
|---|---|---|---|---|
| Student model 1 1M param - 4 MB 84.5% - 88.8% | PM Quant.(No bucket) | 9.30 % | 67.99 % | 88.91 % |
| | PM Quant. (with bucket) | 10.53 % | 87.18 % | 88.80 % |
| | Quantized Distill. | 82.4 % | 88.00 % | 88.82 % |
| | Differentiable Quant. | 80.43% | 88.31 % | —— |
| Student model 2 0.3M param - 1.27 MB 80.3% - 84.3% | PM Quant. (No bucket) | 10.15 % | 68.05 % | 84.38 % |
| | PM Quant. (with bucket) | 11.89 % | 81.96 % | 84.38 % |
| | Quantized Distill. | 74.22 % | 83.92 % | 84.22 % |
| | Differentiable Quant. | 72.79 % | 83.49 % | —— |
| Student model 3 0.1M param - 0.45 MB 71.6% - 78.2% | PM Quant. (No bucket) | 10.15 % | 61.30 % | 78.04 % |
| | PM Quant. (with bucket) | 10.38 % | 72.44 % | 78.10 % |
| | Quantized Distill. | 67.02 % | 77.75 % | 77.92 % |
| | Differentiable Quant. | 57.84 % | 77.36 % | —— |

Table 2: CIFAR10 accuracy. Teacher model: 5.3M param, 21 MB, accuracy 89.71%. Details about the resulting size of the models are reported in table 14 in the appendix.

|  |  | 2 bits | 4 bits |
|---|---|---|---|
| Deeper student 5.8M param - 23.2 MB 93.22% - 92.6% | PM Quant.(No bucket) | 12.60 % | 91.11 % |
| | PM (with bucketing) | 45.82 % | 92.30 % |
| | Quantized Distilled | 89.33 % | 92.17 % |

Table 3: CIFAR10 accuracy (Wide Residual Network). Teacher model: 145M param, 580 MB, accuracy 95.7 %. Details about the resulting size of the models are reported in table 17 in the appendix.

| Student model |  | 2 bits | 4 bits |
|---|---|---|---|
| 82.7M param - 330 MB 95.3% - 94.19% | PM Quant. (with bucket) | 15.35 % | 81.1 % |
| | Quantized Distill. | 94.23 % | 94.73 % |

but it has much faster convergence. Further, we highlight the good accuracy of the much simpler PM quantization method with bucketing at higher bit width (4 and 8 bits).

**CIFAR-100 Experiments.** Next, we perform image classification with the full 100 classes. Here, we focus on 2bit and 4bit quantization, and on a single student architecture. The baseline architecture is a wide residual network with 28 layers, and 36.5M parameters, which is state-of-the-art for its depth on this dataset. The student has depth and width reduced by $20\%$, and half the parameters. It is chosen so that reaches the same accuracy as the teacher model when distilled at full precision. Accuracy results are given in Table 4. More details are reported in Table 20, in the appendix.

Table 4: CIFAR100 accuracy and model size. Teacher: 36.5M param, 146 MB, acc. 77.21 %.

|  |  | 2 bits | 4 bits |
|---|---|---|---|
| Student model 17.2M param - 68.8 MB 77.08% - 77.24% | PM Quant. (No bucket) | 1.38 % - 3.18 MB | 1.29 % - 5.77 MB |
| | PM Quant. (with bucket) | 1.00 % - 3.9 MB | 73.5 % - 8.2 MB |
| | Quantized Distill. | 27.84 % - 4.3 MB | 76.31 % - 8.2 MB |
| | Differentiable Quant. | 49.32 % - 7.9 MB | 77.07 % - 12.4 MB |

The results confirm the trend from the previous dataset, with distilled and differential quantization preserving accuracy within less than 1% at 4bit precision. However, we note that accuracy loss is catastrophic at 2bit precision, probably because of reduced model capacity. We note that differentiable quantization is able to best recover accuracy for this harder task.

**OpenNMT Experiments.** The OpenNMT integration test dataset (Ope) consists of 200K train sentences and 10K test sentences for a German-English translation task. To train and test models we

use the OpenNMT PyTorch codebase (Klein et al., 2017). We modified the code, in particular by adding the quantization algorithms and the distillation loss. As measure of fit we will use perplexity and the BLEU score, the latter computed using the `multi-bleu.perl` code from the moses project (mos).

Our target models consist of an embedding layer, an encoder consisting of $n$ layers of LSTM, a decoder consisting of $n$ layers of LSTM, and a linear layer. The decoder also uses the global attention mechanism described in Luong et al. (2015). For the teacher network we set $n = 2$, for a total of 4 LSTM layers with LSTM size 500. For the student networks we choose $n = 1$, for a total of 2 LSTM layers. We vary the LSTM size of the student networks and for each one, we compute the distilled model and the quantized versions for varying bit width. Results are summarized in Table 5. The BLEU scores below the student model refer to the BLEU scores of the normal and distilled model respectively (trained with full precision). Details about the resulting size of the models are reported in table 23 in the appendix.

Table 5: OpenNMT dataset BLEU score and perplexity (ppl). Teacher model: 84.8M param, 340 MB, 26.1 ppl, 15.88 BLEU. Details about the resulting size of the models are reported in table 23 in the appendix.

| | | 2 bits | 4 bits |
|---|---|---|---|
| Student model 1 | PM Quant.(No bucket) | $0.00 - 2 \cdot 10^{17}$ ppl | $0.24 - 2 \cdot 10^6$ ppl |
| 81.6M param - 326 MB | PM Quant. (with bucket) | 4.12 - 125.1 ppl | 16.29 - 26.2 ppl |
| 14.97 - 16.13 BLEU | Quantized Distill. | 0.00 - 6645 ppl | 15.73 - 25.43 ppl |
| | Differentiable Quant. | 0.7 - 249 ppl | 15.01 - 28.8 ppl |
| Student model 2 | PM Quant. (No bucket) | $0.00 - 5 \cdot 10^8$ ppl | 6.65 - 71.78 ppl |
| 64.8M param - 249 MB | PM Quant. (with bucket) | 1.72 - 286.98 ppl | 15.19 - 28.95 ppl |
| 14.22 - 15.48 BLEU | Quantized Distill. | 0.00 - 4035 ppl | 15.26 - 29.1 ppl |
| | Differentiable Quant. | 0.28 - 306 ppl | 13.86 - 31.33 ppl |
| Student model 3 | PM Quant. (No bucket) | $0.00 - 3 \cdot 10^8$ ppl | 5.47 - 106.5 ppl |
| 57.2M param - 228 MB | PM Quant. (with bucket) | 0.24 - 1984 ppl | 12.64 - 36.56 ppl |
| 12.45 - 13.8 BLEU | Quantized Distill. | 0.14 - 731 ppl | 12 - 37 ppl |
| | Differentiable Quant. | 0.26 - 306 ppl | 12.06 - 38.44 ppl |

A reasonable intuition would be that recurrent neural networks should be harder to quantize than convolutional neural networks, as quantization errors do not *average out* when executing repeatedly through the same cell, but accumulate. Results contradict this intuition. In particular, medium and large-sized students are able to essentially recover the same scores as the teacher model on this dataset. Perhaps surprisingly, bucketing PM and quantized distillation perform equally well for 4bit quantization. As expected, cell size is an important indicator for accuracy, although halving both cell size and the number of layers can be done without significant loss.

## 6.2 LARGER DATASETS

**WMT13 Experiments.** We run a similar LSTM architecture as above for the WMT13 dataset (Koehn, 2005) (1.7M sentences train, 190K sentences test) and we provide additional experiments for quantized distillation technique, see Table 6. We note that, on this large dataset, PM quantization does not perform well, even with bucketing. On the other hand, quantized distillation with 4bits of precision has *higher* BLEU score than the teacher, and similar perplexity.

Table 6: WMT13 dataset BLEU score and perplexity (ppl). Teacher model: 84.8M param, 340 MB, 5.8 ppl, 34.7 BLEU. Details about model size are reported in Table 26.

| | | 4 bits |
|---|---|---|
| Student Model | PM Quant. (No bucket) | 21.38 BLEU - 12.61 ppl |
| 81.6M param - 326 MB | PM Quant. (with bucket) | 27.73 BLEU - 7.4 ppl |
| 30.22 - 30.21 BLEU | Quantized Distill. | 35.32 BLEU - 6.48 ppl |

**The ImageNet Dataset.** We also experiment with ImageNet using the ResNet architecture (He et al., 2016a). In the first experiment, we use a ResNet34 teacher, and a student ResNet18 student model. Experiments quantizing the standard version of this student resulted in an accuracy loss of around

4%, and hence we experiment with a wider model, which doubles the number of filters for each convolutional layer. We call this 2xResNet18. This is in line with previous work on wide ResNet architectures (Zagoruyko & Komodakis, 2016), wide students for distillation (Ba & Caruana, 2013), and wider quantized networks (Mishra et al., 2017). We also note that, in line with previous work on this dataset (Zhu et al., 2016; Mishra et al., 2017), we do not quantize the first and last layers of the models, as this can hurt accuracy.

After 62 epochs of training, the quantized distilled 2xResNet18 with 4 bits reaches a *validation* accuracy of 73.31%. Surprisingly, this is higher than the *unquantized* ResNet18 model (69.75%), and has virtually the same accuracy as the ResNet34 teacher. In terms of size, this model is more than $2\times$ smaller than ResNet18 (but has higher accuracy), and is $4\times$ smaller than ResNet34, and about $1.5\times$ faster on inference, as it has fewer layers. This is state-of-the-art for 4bit models with 18 layers; to our knowledge, no such model has been able to surpass the accuracy of ResNet18.

We re-iterated this experiment using a 4-bit quantized 2xResNet34 student transferring from a ResNet50 full-precision teacher. We obtain a 4-bit quantized student of almost the same accuracy, which is 50% shallower and has a $2.5\times$ smaller size.

Table 7: Imagenet accuracy and model size. Bucket size = 256.

| Model name | Top-1 Accuracy | Top-5 Accuracy | # of parameters | Size (MB) |
|---|---|---|---|---|
| Teacher model: ResNet34 | 73.31 % | 91.42 % | 21.79 millions | 87.16 MB |
| ResNet18 normal | 69.75 % | 89.07 % | 11.69 millions | 46.76 MB |
| 2xResNet18 QD 4 bits | 73.10 % | 91.17 % | 45.69 millions | 21.98 MB |
| Teacher model: ResNet50 | 76.13 % | 92.86 % | 25.55 millions | 102.2 MB |
| 2xResNet34 QD 4 bits | 76.07 % | 92.71 % | 86.11 millions | 41.53 MB |

## 6.3 ADDITIONAL EXPERIMENTS

**Distillation Loss versus Normal Loss.** One key question we are interested in is whether distillation loss is a consistently better metric when quantizing, compared to standard loss. We tested this for CIFAR-10, comparing the performance of quantized training with respect to each loss. At 2bit precision, the student converges to 67.22% accuracy with normal loss, and to 82.40% with distillation loss. At 4bit precision, the student converges to 86.01% accuracy with normal loss, and to 88.00% with distillation loss. On OpenNMT, we observe a similar gap: the 4bit quantized student converges to 32.67 perplexity and 15.03 BLEU when trained with normal loss, and to 25.43 perplexity (better than the teacher) and 15.73 BLEU when trained with distillation loss. This strongly suggests that *distillation loss is superior* when quantizing. For details, see Section A.4.1 in the Appendix.

**Impact of Heuristics on Differentiable Quantization.** We also performed an in-depth study of how the various heuristics impact accuracy. We found that, for differentiable quantization, redistributing bits according to the gradient norm of the layers is absolutely essential for good accuracy; quantiles and distillation loss also seem to provide an improvement, albeit smaller. Due to space constraints, we defer the results and their discussion to Section A.4.2 of the Appendix.

**Inference Speed.** In general, shallower students lead to an almost-linear decrease in inference cost, w.r.t. the depth reduction. For instance, in the CIFAR-10 experiments with the wide ResNet models, the teacher forward pass takes 67.4 seconds, while the student takes 43.7 seconds; roughly a 1.5x speedup, for 1.75x reduction in depth. On the ImageNet test set using 4 GPUs (data-parallel), a forward pass takes 263 seconds for ResNet34, 169 seconds for ResNet18, and 169 seconds for our 2xResNet18. (So, while having more parameters than ResNet18, it has the same speed because it has the same number of layers, and is not wide enough to saturate the GPU. We note that we did not exploit 4bit weights, due to the lack of hardware support.) Inference on our model is 1.5 times faster, while being 1.8 times shallower, so here the speedup is again almost linear.

## 7 DISCUSSION

We have examined the impact of combining distillation and quantization when compressing deep neural networks. Our main finding is that, when quantizing, one can (and should) leverage large, accurate models via distillation loss, if such models are available. We have given two methods to do just that, namely quantized distillation, and differentiable quantization. The former acts directly

on the training process of the student model, while the latter provides a way of optimizing the quantization of the student so as to best fit the teacher model.

Our experimental results suggest that these methods can compress existing models by up to an order of magnitude in terms of size, on small image classification and NMT tasks, while preserving accuracy. At the same time, we note that distillation also provides an automatic improvement in *inference speed*, since it generates shallower models. One of our more surprising findings is that naive uniform quantization *with bucketing* appears to perform well in a wide range of scenarios. Our analysis in Section 2.2 suggests that this may be because bucketing provides a way to parametrize the Gaussian-like noise induced by quantization. Given its simplicity, it could be used consistently as a baseline method.

In our experimental results, we performed manual architecture search for the depth and bit width of the student model, which is time-consuming and error-prone. In future work, we plan to examine the potential of reinforcement learning or evolution strategies to discover the structure of the student for best performance given a set of space and latency constraints. The second, and more immediate direction, is to examine the practical speedup potential of these methods, and use them together and in conjunction with existing compression methods such as weight sharing Han et al. (2015) and with existing low-precision computation frameworks, such as NVIDIA TensorRT, or FPGA platforms.

ACKNOWLEDGEMENTS

We would like to thank Ce Zhang (ETH Zúrich), Hantian Zhang (ETH Zúrich) and Martin Jaggi (EPFL) for their support with experiments and valuable feedback.

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

## A    FULL EXPERIMENTAL RESULTS

### A.1    CIFAR10

The model used to train CIFAR10 is the one described in Urban et al. (2016) with some minor modifications. We use standard data augmentation techniques, including random cropping and random flipping. The learning rate schedule follows the one detailed in the paper. The structure of the models we experiment with consists of some convolutional layers, mixed with dropout layers and max pooling layers, followed by one or more linear layers.

The model used are defined in Table 8. The $c$ indicates a convolutional layer, mp a max pooling layer, dp a dropout layer, fc a linear (fully connected) layer. The exponent indicates how many consecutive layers of the same type are there, while the number in front of the letter determines the size of the layer. In the case of convolutional layers is the number of filters. All convolutional layers of the teacher are 3x3, while the convolutional layers in the smaller models are 5x5.

Table 8: CIFAR10: model specifications

| Teacher model | $76c^2$-mp-dp-$126c^2$-mp-dp-$148c^4$-mp-dp-1200fc-dp-1200fc |
|---|---|
| Smaller model 1 | $75c$-mp-dp-$50c^2$-mp-dp-$25c$-mp-dp-500fc-dp |
| Smaller model 2 | $50c$-mp-dp-$25c^2$-mp-dp-$10c$-mp-dp-400fc-dp |
| Smaller model 3 | $25c$-mp-dp-$10c^2$-mp-dp-$5c$-mp-dp-300fc-dp |

Following the authors of the paper, we don't use dropout layers when training the models using distillation loss. Distillation loss is computed with a temperature of $T = 5$.

Table 9 reports the accuracy of the models trained (in full precision) and their size. Table 10 reports the accuracy achieved with each method, and table 11 reports the optimal mean bit length using Huffman encoding and resulting model size.

Table 9: CIFAR10: Teacher and distilled model accuracy, full precision

| Model name | Test accuracy | # of parameters | Size (MB) |
|---|---|---|---|
| Teacher model | 89.7 % | 5.3 millions | 21.3 MB |
| Smaller model 1 | 84.5 % | 1.0 millions | 4.00 MB |
| Distilled model 1 | 88.8 % | 1.0 millions | 4.00 MB |
| Smaller model 2 | 80.3 % | 0.3 millions | 1.27 MB |
| Distilled model 2 | 84.3 % | 0.3 millions | 1.27 MB |
| Smaller model 3 | 71.6 % | 0.1 millions | 0.45 MB |
| Distilled model 3 | 78.2 % | 0.1 millions | 0.45 MB |

Table 10: CIFAR10: Test accuracy for quantized models. Results computed with bucket size = 256

| | 2 bits | 4 bits | 8 bits |
|---|---|---|---|
| PM Quant. 1 (No bucket) | 9.30 % | 67.99 % | 88.91 % |
| PM Quant. 1 (with bucket) | 10.53 % | 87.18 % | 88.80 % |
| Quantized Distill. 1 | 82.4 % | 88.00 % | 88.82 % |
| Differentiable Quant. 1 | 80.43% | 88.31 % | — |
| PM Quant. 2 (No bucket) | 10.15 % | 68.05 % | 84.38 % |
| PM Quant. 2 (with bucket) | 11.89 % | 81.96 % | 84.38 % |
| Quantized Distill. 2 | 74.22 % | 83.92 % | 84.22 % |
| Differentiable Quant. 2 | 72.79 % | 83.49 % | — |
| PM Quant. 3 (No bucket) | 10.15 % | 61.30 % | 78.04 % |
| PM Quant. 3 (with bucket) | 10.38 % | 72.44 % | 78.10 % |
| Quantized Distill. 3 | 67.02 % | 77.75 % | 77.92 % |
| Differentiable Quant. 3 | 57.84 % | 77.36 % | — |

We also performed an experiment with a deeper student model. The architecture is $76c^3$-mp-dp-$126c^3$-mp-dp-$148c^5$-mp-dp-1000fc-dp-1000fc-dp-1000fc (following the same notation as in table 8). We use the same teacher as in the previous experiments. Results are in table 13.

### A.1.1 CIFAR10 - WIDERESNET ARCHITECTURE

For our second set of experiments on CIFAR10 with the WideResNet architecture, see table 15. Note that we increase the number of filters but reduce the depth of the model. The implementation of WideResNet used can be found on GitHub [2]. Results of quantized methods are in table 16 while the size of the resulting models is detailed in table 17.

---

[2]https://github.com/meliketoy/wide-resnet.pytorch

Table 11: CIFAR10: Optimal length Huffman encoding and resulting model size. Bucket size = 256

|  | 2 bits | 4 bits | 8 bits |
|---|---|---|---|
| PM Quant. 1 (No bucket) | 1.34 bits - 0.17 MB | 2.43 bits - 0.3 MB | 6.48 bits - 0.81 MB |
| PM Quant. 1 (with bucket) | 1.58 bits - 0.22 MB | 3.52 - 0.47 MB | 7.58 bits - 0.98 MB |
| Quantized Distill. 1 | 1.7 bits - 0.24 MB | 3.64 bits - 0.48 MB | 7.70 bits - 1 MB |
| Differentiable Quant. 1 | 3.18 bits - 0.43 MB | 5.34 bits - 0.7 MB | —— |
| | | | |
| PM Quant. 2 (No bucket) | 1.43 bits - 0.05 MB | 2.6 bits - 0.1 MB | 6.65 bits - 0.26 MB |
| PM Quant. 2 (with bucket) | 1.6 bits - 0.07 MB | 3.58 bits - 0.15 MB | 7.64 bits - 0.31 MB |
| Quantized Distill. 2 | 1.7 bits - 0.08 MB | 3.55 bits - 0.15 MB | 7.64 bits - 0.31 MB |
| Differentiable Quant. 2 | 3.16 bits - 0.13 MB | 5.34 bits - 0.22 MB | —— |
| | | | |
| PM Quant. 3 (No bucket) | 1.46 bits - 0.02 MB | 2.62 bits - 0.03 MB | 6.66 bits - 0.09 MB |
| PM Quant. 3 (with bucket) | 1.58 bits - 0.026 MB | 3.51 bits - 0.053 MB | 7.56 bits - 0.1 MB |
| Quantized Distill. 3 | 1.64 bits - 0.027 MB | 3.53 bits - 0.053 MB | 7.59 bits - 0.11 MB |
| Differentiable Quant. 3 | 3.12 bits - 0.04 MB | 5.41 bits - 0.08 MB | —— |

Table 12: CIFAR10: Teacher and distilled model accuracy, full precision

| Model name | Test accuracy | # of parameters | Size (MB) |
|---|---|---|---|
| Teacher model | 89.7 % | 5.3 millions | 21.3 MB |
| | | | |
| Deeper normal model 1 | 93.22 % | 5.8 millions | 23.40 MB |
| Deeper distilled model 1 | 92.60 % | 5.8 millions | 23.40 MB |

Table 13: CIFAR10: Test accuracy for quantized deeper student model. Results computed with bucket size = 256

|  | 2 bits | 4 bits |
|---|---|---|
| PM Quant. (No bucket) | 12.60 % | 91.11 % |
| PM Quant. (with bucket) | 45.82 % | 92.30 % |
| Quantized Distill. | 89.33 % | 92.17 % |

Table 14: CIFAR10: Optimal length Huffman encoding and resulting model size for deeper student model. Bucket size = 256

|  | 2 bits | 4 bits |
|---|---|---|
| PM Quant. (No bucket) | 1.50 bits - 1.13 MB | 3.21 bits - 2.38 MB |
| PM Quant. (with bucket) | 1.82 bits - 1.55 MB | 3.92 bits - 3.07 MB |
| Quantized Distill. | 1.84 bits - 1.56 MB | 3.92 bits - 3.08 MB |

Table 15: CIFAR10: Teacher and distilled model accuracy, full precision, wide resnet

| Model name | Structure | Test accuracy | # of parameters | Size (MB) |
|---|---|---|---|---|
| Teacher model | depth = 28, wide_factor = 20 | 95.74 % | 145 millions | 580 MB |
| Smaller model | depth = 22, wide_factor = 16 | 95.19 % | 82.7 millions | 330 MB |
| Distilled model | depth = 22, wide_factor = 16 | 94.19 % | 82.7 millions | 330 MB |

## A.2 CIFAR100

For our CIFAR100 experiments, we use the same implementation of wide residual networks as in our CIFAR10 experiments. The wide factor is a multiplicative factor controlling the amount of filters in each layer; for more details please refer to the original paper Zagoruyko & Komodakis (2016). We train for 200 epochs with an initial learning rate of 0.1.

Table 16: CIFAR10: Test accuracy for quantized models. Results computed with bucket size = 256

|  | 2 bits | 4 bits |
|---|---|---|
| PM Quant. (with bucket) | 15.35 % | 81.09 % |
| Quantized Distill. | 94.23 % | 94.73 % |

Table 17: CIFAR10: Optimal length Huffman encoding and resulting model size. Bucket size = 256

|  | 2 bits | 4 bits |
|---|---|---|
| PM Quant. (with bucket) | 1.44 bits - 17.56 MB | 2.62 bits - 29.75 MB |
| Quantized Distill. | 1.54 bits - 17.81 MB | 3.48 bits - 38.65 MB |

For the CIFAR100 experiments we focused on one student model. Distillation loss is computed with a temperature of $T = 5$.

Table 18: CIFAR100: Teacher and distilled model accuracy, full precision

| Model name | Structure | Test accuracy | # of parameters | Size (MB) |
|---|---|---|---|---|
| Teacher model | depth = 28, wide_factor = 10 | 77.21 % | 36.5 millions | 146 MB |
| Smaller model | depth = 22, wide_factor = 8 | 77.08 % | 17.2 millions | 68.8 MB |
| Distilled model | depth = 22, wide_factor = 8 | 77.24 % | 17.2 millions | 68.8 MB |

Table 19: CIFAR100: Test accuracy for quantized models. Results computed with bucket size = 256

|  | 2 bits | 4 bits |
|---|---|---|
| PM Quant. (No bucket) | 1.38% | 1.29% |
| PM Quant. (with bucket) | 1.00 % | 73.47% |
| Quantized Distill. | 27.84% | 76.31% |
| Differentiable Quant. | 49.32% | 77.07% |

Table 20: CIFAR100: Optimal length Huffman encoding and resulting model size. Bucket size = 256

|  | 2 bits | 4 bits |
|---|---|---|
| PM Quant. (No bucket) | 1.47 bits - 3.18 MB | 2.68 bits - 5.77 MB |
| PM Quant. (with bucket) | 1.56 bits - 3.90 MB | 3.55 bits - 8.18 MB |
| Quantized Distill. | 1.73 bits - 4.27 MB | 3.54 bits - 8.16 MB |
| Differentiable Quant. | 3.23 bits - 7.84 MB | 5.53 bits - 12.44 MB |

### A.3 OPENTNMT INTEGRATION TEST DATASET

As mentioned in the main text, we use the openNMT-py codebase. We slightly modify it to add distillation loss and the quantization methods proposed. We mostly use standard options to train the model; in particular, the learning rate starts at 1 and is halved every epoch starting from the first epoch where perplexity doesn't drop on the test set. We train every model for 15 epochs. Distillation loss is computed with a temperature of $T = 1$.

### A.4 WMT13 DATASET

For the WMT13 datasets, we run a similar architecture. We ran all models for 15 epochs; the smaller model overfit with 15 epochs, so we ran it for 5 epochs instead.

Table 21: openNMT integ: Teacher and distilled models perplexity and BLEU, full precision

| Model name | Structure | Perplexity | BLEU | # of parameters | Size (MB) |
|---|---|---|---|---|---|
| Teacher model | 4 LSTM layer, 500 cell size | 26.21 | 15.88 | 84.8 millions | 339.28 MB |
| | | | | | |
| Smaller model 1 | 2 LSTM layer, 512 cell size | 33.03 | 14.97 | 81.6 millions | 326.57 MB |
| Distilled model 1 | 2 LSTM layer, 512 cell size | 25.55 | 16.13 | 81.6 millions | 326.57 MB |
| | | | | | |
| Smaller model 2 | 2 LSTM layer, 256 cell size | 34.5 | 14.22 | 64.8 millions | 249.56 MB |
| Distilled model 2 | 2 LSTM layer, 256 cell size | 27.7 | 15.48 | 64.8 millions | 249.56 MB |
| | | | | | |
| Smaller model 3 | 2 LSTM layer, 128 cell size | 39.5 | 12.45 | 57.2 millions | 228.85 MB |
| Distilled model 3 | 2 LSTM layer, 128 cell size | 33.78 | 13.8 | 57.2 millions | 228.85 MB |

Table 22: openNMT integ: Test accuracy for quantized models. Results computed with bucket size = 256

| | 2 bits | 4 bits |
|---|---|---|
| PM Quant. 1 (No bucket) | $2 \cdot 10^{17}$ ppl - 0.00 BLEU | $2.7 \cdot 10^6$ ppl - 0.24 BLEU |
| PM Quant. 1 (with bucket) | 125.1 ppl - 4.12 BLEU | 26.21 ppl - 16.29 BLEU |
| Quantized Distill. 1 | 6645 ppl - 0.00 BLEU | 25.43 ppl - 15.73 BLEU |
| Differentiable Quant. 1 | 249 ppl - 0.7 BLEU | 28.8 ppl - 15.01 BLEU |
| | | |
| PM Quant. 2 (No bucket) | $5 \cdot 10^8$ ppl - 0.00 BLEU | 71.78 ppl - 6.65 BLEU |
| PM Quant. 2 (with bucket) | 286.98 ppl - 1.72 BLEU | 28.95 ppl - 15.19 BLEU |
| Quantized Distill. 2 | 4035 ppl - 0.00 BLEU | 29.1 ppl - 15.26 BLEU |
| Differentiable Quant. 2 | 306 ppl - 0.28 BLEU | 31.33 ppl - 13.86 BLEU |
| | | |
| PM Quant. 3 (No bucket) | $3 \cdot 10^8$ ppl - 0.00 BLEU | 106.5 ppl - 5.47 BLEU |
| PM Quant. 3 (with bucket) | 1984 ppl - 0.24 BLEU | 36.56 ppl - 12.64 BLEU |
| Quantized Distill. 3 | 731 ppl - 0.14 BLEU | 37 ppl - 12 BLEU |
| Differentiable Quant. 3 | 306 ppl - 0.26 BLEU | 38.4 ppl - 12.06 BLEU |

Table 23: openNMT integ: Optimal length Huffman encoding and resulting model size. Bucket size = 256

| | 2 bits | 4 bits |
|---|---|---|
| PM Quant. 1 (No bucket) | 1.36 bits - 13.93 MB | 1.77 bits - 18.10 MB |
| PM Quant. 1 (with bucket) | 1.65 bits - 19.47 MB | 3.69 bits - 40.26 MB |
| Quantized Distill. 1 | 1.75 bits - 20.4 MB | 3.66 bits - 39.97 MB |
| Differentiable Quant. 1 | 1.72 bits - 20.1 MB | 4.38 bits - 47.32 MB |
| | | |
| PM Quant. 2 (No bucket) | 1.34 bits - 10.89 MB | 1.86 bits - 15.09 MB |
| PM Quant. 2 (with bucket) | 1.65 bits - 15.4 MB | 3.68 bits - 31.91 MB |
| Quantized Distill. 2 | 1.85 bits - 17.05 MB | 3.68 bits - 31.91 MB |
| Differentiable Quant. 2 | 1.93 bits - 17.67 MB | 4.17 bits - 35.83 MB |
| | | |
| PM Quant. 3 (No bucket) | 1.47 bits - 10.54 MB | 2.13 bits - 15.24 MB |
| PM Quant. 3 (with bucket) | 1.65 bits - 13.6 MB | 3.68 bits - 28.14 MB |
| Quantized Distill. 3 | 1.86 bits - 15.13 MB | 3.68 bits - 28.18 MB |
| Differentiable Quant. 3 | 1.99 bits - 16.04 MB | 4.25 bits - 31.18 MB |

### A.4.1 DISTILLATION VERSUS STANDARD LOSS FOR QUANTIZATION

In this section we highlight the positive effects of using distillation loss during quantization. We take models with the same architecture and we train them with the same number of bits; one of the models is trained with normal loss, the other with the distillation loss with equal weighting between soft cross entropy and normal cross entropy (that is, it is the quantized distilled model).

Table 24: WMT13: Teacher and distilled models perplexity and BLEU, full precision

| Model name | Structure | Perplexity | BLEU | # of parameters | Size (MB) |
|---|---|---|---|---|---|
| Teacher model | 4 LSTM layer, 500 cell size | 5.83 | 34.77 | 84.8 millions | 339.28 MB |
| Smaller model 1 | 2 LSTM layer, 500 size | 7.98 | 30.22 | 80.8 millions | 323.25 MB |
| Distilled model 1 | 2 LSTM layer, 550 cell size | 7.18 | 30.21 | 84.3 millions | 337.21 MB |

Table 25: WMT13: Test accuracy for quantized models. Results computed with bucket size = 256

| | 4 bits |
|---|---|
| PM Quant. (No bucket) | 12.17 ppl - 22.79 BLEU |
| PM Quant. (with bucket) | 7.34 ppl - 26.18 BLEU |
| Quantized Distill. | 6.48 ppl - 35.32 BLEU |

Table 26: WMT13: Optimal length Huffman encoding and resulting model size. Bucket size = 256

| | 4 bits |
|---|---|
| PM Quant. 1 (No bucket) | 1.98 bits - 20.92 MB |
| PM Quant. 1 (with bucket) | 3.63 bits - 41.02 MB |
| Quantized Distill. 1 | 3.65 bits - 41.16 MB |

Table 27 shows the results on the CIFAR10 dataset; the models we train have the same structure as the Smaller model 1, see Section A.1.

Table 28 shows the results on the openNMT integration test dataset; the models trained have the same structure of Smaller model 1, see Section A.3. Notice that distillation loss can significantly improve the accuracy of the quantized models.

Table 27: CIFAR10: Distillation loss vs normal loss when quantizing

| | 2 bits | 4 bits |
|---|---|---|
| Normal loss | 67.22 % | 86.01 % |
| Distillation loss | **82.40** % | **88.00** % |

Table 28: openNMT integ: Distillation loss vs normal loss when quantizing

| | 4 bits | |
|---|---|---|
| Normal loss | 32.67 ppl | 15.03 BLEU |
| Distillation loss | **25.43 ppl** | **15.73 BLEU** |

These results suggest that quantization works better when combined with distillation, and that we should try to take advantage of this whenever we are quantizing a neural network.

### A.4.2 DIFFERENT HEURISTICS FOR DIFFERENTIABLE QUANTIZATION

To test the different heuristics presented in Section 4.2, we train with differentiable quantization the Smaller model 1 architecture specified in Section A.1 on the cifar10 dataset. The same model is trained with different heuristics to provide a sense of how important they are; the experiments is performed with 2 and 4 bits.

Results suggests that when using 4 bits, the method is robust and works regardless. When using 2 bits, redistributing bits according to the gradient norm of the layers is absolutely essential for this method to work ; quantiles starting point also seem to provide an small improvement, while using distillation loss in this case does not seem to be crucial.

Table 29: Results with automatically redistributed bits

|        |                   | Quantiles | Uniform  |
|--------|-------------------|-----------|----------|
| 2 bits | Distillation loss | 82.94 %   | 78.76 %  |
|        | Normal loss       | 83.67 %   | 76.60 %  |
| 4 bits | Distillation loss | 88.93 %   | 88.50 %  |
|        | Normal loss       | 88.80 %   | 88.74 %  |

Table 30: Results without automatically redistributed bits

|        |                   | Quantiles | Uniform  |
|--------|-------------------|-----------|----------|
| 2 bits | Distillation loss | 19.69 %   | 22.81 %  |
|        | Normal loss       | 25.28 %   | 22.11 %  |
| 4 bits | Distillation loss | 88.39 %   | 88.67 %  |
|        | Normal loss       | 88.43 %   | 88.44 %  |

# B  QUANTIZATION IS EQUIVALENT TO ASYMPTOTICALLY NORMALLY DISTRIBUTED NOISE

In this section we will prove some results about the uniform quantization function, including the fact that is asymptotically normally distributed, see subsection B.1 below. Clearly, we refer to the stochastic version, see Section 2.1.

**Unbiasedness**

We first start proving the unbiasedness of $\hat{Q}$;

$$E[\hat{Q}(\hat{v})_i] = \frac{\lfloor \hat{v}_i s \rfloor}{s} + \frac{1}{s}E[\xi_i] = \frac{\lfloor \hat{v}_i s \rfloor}{s} + \frac{1}{s}(s\hat{v}_i - \lfloor \hat{v}_i s \rfloor) = \hat{v}_i \tag{8}$$

Then it is immediate that

$$E[Q(v)] = \alpha E\left[\hat{Q}\left(\frac{v-\beta}{\alpha}\right)\right] + \beta = \alpha \frac{v-\beta}{\alpha} + \beta = v \tag{9}$$

**Bounds on second and third moment**

We will write out bounds on $\hat{Q}$; the analogous bounds on $Q$ are then straightforward. For convenience, let us call $\hat{l}_i = \lfloor \hat{v}_i s \rfloor$

$$E[\hat{Q}(\hat{v})_i^2] = \frac{\hat{l}_i^2}{s^2} + \frac{1}{s^2}E[\xi_i^2] + 2\frac{\hat{l}_i}{s^2}E[\xi_i] = \tag{10}$$

$$= \frac{\hat{l}_i^2}{s^2} + \frac{1}{s^2}(s\hat{v}_i - \hat{l}_i) + 2\frac{\hat{l}_i}{s^2}(s\hat{v}_i - \hat{l}_i) = \tag{11}$$

$$= \frac{1}{s^2}\left[\hat{v}_i s(1 + 2\hat{l}_i) - \hat{l}_i(\hat{l}_i + 1)\right] \tag{12}$$

And given that $\hat{l}_i \le \hat{v}_i s \le \hat{l}_i + 1$, we readily find

$$\frac{\hat{l}_i^2}{s^2} \le E[\hat{Q}(\hat{v})_i^2] \le \frac{(\hat{l}_i + 1)^2}{s^2} \tag{13}$$

For the third moment, we have

$$E[\hat{Q}(\hat{v})_i^3] = \frac{\hat{l}_i^3}{s^3} + \frac{1}{s^3}E[\xi_i^3] + 3\frac{\hat{l}_i}{s^3}E[\xi_i^2] + 3\frac{\hat{l}_i^2}{s^3}E[\xi_i] = \tag{14}$$

$$= \frac{\hat{l}_i^3}{s^3} + \frac{1}{s^3}(s\hat{v}_i - \hat{l}_i) + 3\frac{\hat{l}_i}{s^3}(s\hat{v}_i - \hat{l}_i) + 3\frac{\hat{l}_i^2}{s^3}(s\hat{v}_i - \hat{l}_i) = \tag{15}$$

$$= \frac{1}{s^3}\left[\hat{v}_i s(3\hat{l}_i^2 + 3\hat{l}_i + 1) - \hat{l}_i(2\hat{l}_i^2 + 3\hat{l}_i + 1)\right] \tag{16}$$

And as before, the bounds are

$$\frac{\hat{l}_i^3}{s^3} \le E[\hat{Q}(\hat{v})_i^3] \le \frac{(\hat{l}_i + 1)^3}{s^3} \tag{17}$$

## B.1 Asymptotic normality

Most of neural networks operations are scalar product computation. Therefore, the scalar product of the quantized weights and the inputs is an important quantity:

$$Q(v)^T x = \sum_{i=1}^{n} Q(v_i)x_i$$

We already know from section B that the quantization function is unbiased; hence we know that

$$\sum_{i=1}^{n} Q(v_i)x_i = \sum_{i=1}^{n} v_i x_i + \varepsilon_n \tag{18}$$

with $\varepsilon_n$ is a zero-mean random variable. We will show that $\varepsilon_n$ tends in distribution to a normal random variable. To prove asymptotic normality, we will use a generalized version of the central limit theorem due to Lyapunov:

**Theorem B.1** (Lyapunov Central Limit Theorem). *Let $\{X_1, X_2, \dots\}$ be a sequence of independent random variables, each with finite expected value $\mu_i$ and variance $\sigma_i^2$. Define $s_n^2 = \sum_{i=1}^{n} \sigma_i^2$. If, for some $\delta > 0$, the Lyapunov condition*

$$\lim_{n\to\infty} \frac{1}{s_n^{2+\delta}} \sum_{i=1}^{n} E\left[|X_i - \mu_i|^{2+\delta}\right] = 0 \tag{19}$$

*is satisfied, then*

$$\frac{1}{s_n} \sum_{i=1}^{n} (X_i - \mu_i) \xrightarrow{D} N(0, 1) \tag{20}$$

We can now state the theorem:

**Theorem B.2.** *Let $v, x$ be two vectors with $n$ elements. Let $Q$ be the uniform quantization function with $s$ levels defined in 2.1 and define $s_n^2 = \sum_{i=1}^{n} Var[Q(v_i)x_i]$. If the elements of $v, x$ are uniformly bounded by $M$ [3] and $\lim_{n\to\infty} s_n = \infty$, then*

$$\sum_{i=1}^{n} Q(v_i)x_i = \sum_{i=1}^{n} v_i x_i + \varepsilon_n \tag{21}$$

*with $E[\varepsilon_n] = 0$ and*

$$\lim_{n\to\infty} \frac{1}{s_n}\varepsilon_n \xrightarrow{D} N(0, 1) \tag{22}$$

---

[3]i.e. there exists a constant $M$ such that for all $n$, $|v_i| \le M$, $|x_i| \le M$ for all $i \in \{1, \dots, n\}$

*Proof.* Using the same notation as theorem B.1, let $X_i = Q(v_i)x_i$, $\mu_i = E[X_i] = v_ix_i$. We already mentioned in 2.1 that these are independent random variables. We will show that the Lyapunov condition holds with $\delta = 1$.

We know that

$$E\left[|X_i - \mu_i|^3\right] = E\left[(X_i - \mu_i)^2|X_i - \mu_i|\right] \leq \frac{M^2}{s} E\left[(X_i - \mu_i)^2\right] \tag{23}$$

In fact,

$$|X_i - \mu_i| = |x_i||Q(v_i) - v_i| = |x_i|\left|\alpha_i\hat{Q}\left(\frac{v_i - \beta_i}{\alpha_i}\right) + \beta_i - v_i\right| \leq \tag{24}$$

$$\leq |x_i|\left|\alpha_i\left(\frac{v_i - \beta_i}{\alpha_i} + \frac{1}{s}\right) + \beta_i - v_i\right| \leq \tag{25}$$

$$\leq |x_i|\frac{M}{s} \leq \frac{M^2}{s} \tag{26}$$

since during quantization we have bins of size $\frac{1}{s}$, so that is the largest error we can make. Also, by hypothesis $M \geq \alpha_i, x_i$ for every $i$.

Hence

$$0 \leq \frac{1}{s_n^3}\sum_{i=1}^{n} E\left[|X_i - \mu_i|^3\right] \leq \frac{1}{s_n^3}\frac{M^2}{s}\sum_{i=1}^{n} E\left[(X_i - \mu_i)^2)\right] = \frac{M^2}{s} \cdot \frac{1}{s_n} \tag{27}$$

and since $\lim_{n\to\infty} s_n = \infty$, we have that the Lyapunov condition is satisfied. Hence

$$\frac{1}{s_n}\sum_{i=1}^{n}(X_i - \mu_i) = \frac{1}{s_n}\sum_{i=1}^{n}(Q(v_i)x_i - v_ix_i) = \frac{1}{s_n}\varepsilon_n \xrightarrow{D} N(0,1) \tag{28}$$

$\square$

**Note about the hypothesis** The two hypothesis that were used to prove the theorem are reasonable and should be satisfied by any practical dataset. Typically we know or we can estimate the range of the values of inputs and weights, so the assumption that they don't get arbitrarily large with $n$ is satisfied. The assumption on the variance is also reasonable; in fact, $s_n^2 = \sum_{i=1}^{n} Var[Q(v_i)x_i]$ consists of a sum of $n$ values. While it is possible for all these values to be 0 (if all $v_i$ are in the form $k/s$, for example, then $s_n^2 = 0$) it is unlikely that a real world dataset would present this characteristic. In fact, it suffices that there exist $\gamma > 0$ and $0 < \delta \leq 1$ such that at least $\delta$-percent of $\sigma_i^2 \geq \gamma$. This implies $s_n^2 \geq \delta\gamma n \to \infty$.

**Asymptitc normality when quantizing inputs** Theorem B.2 can be easily extended to the case when also $x_i$ are quantized. The proof is almost identical; we simply have to set $X_i = Q(v_i)Q(x_i)$, use the independence of $Q(x_i)$ and the fact that $Q(v_i)$ and $|Q(v_i)Q(x_i) - x_iv_i| \leq M^2$. For completeness, we report the statement of the theorem :

**Theorem B.3.** *Let $v, x$ be two vectors with $n$ elements. Let $Q$ be the uniform quantization function with $s$ levels defined in 2.1 and define $s_n^2 = \sum_{i=1}^{n} Var[Q(v_i)Q(x_i)]$. If the elements of $v, x$ are uniformly bounded by $M$ and $\lim_{n\to\infty} s_n = \infty$, then*

$$\sum_{i=1}^{n} Q(v_i)Q(x_i) = \sum_{i=1}^{n} v_ix_i + \varepsilon_n \tag{29}$$

*with $E[\varepsilon_n] = 0$ and*

$$\lim_{n\to\infty} \frac{1}{s_n}\varepsilon_n \xrightarrow{D} N(0,1) \tag{30}$$

