# OpenReview forum: "Model compression via distillation and quantization"
_ICLR.cc/2018/Conference — Accept (Poster)_

### Official Review · AnonReviewer1 · 2017-11-26
**The authors try to compress models by combining distillation and quantization. The problem is important. The solution seems to work on large datasets and large models**

**Rating:** 6
**Confidence:** 4

**Review:**

This paper proposes to learn small and low-cost models by combining distillation and quantization. Two strategies are proposed and the ideas are reasonable and clearly introduced. Experiments on various datasets are conducted to show the effectiveness of the proposed method.

Pros:
(1) The paper is well written, the review of distillation and quantization is clear.
(2) Extensive experiments on vision and neural machine translation are conducted.
(3) Detailed discussions about implementations are provided.

Cons:
(1) The differentiable quantization strategy seems not to be consistently better than the straightforward quantized distillation which may need more research.
(2) The actual speedup is not clearly calculated. The authors claim that the inference times of 2xResNet18 and ResNet18 are similar which seems to be unreasonable. And it seems to need a lot of more work to make the idea really practical.

Finally, I am curious whether the idea will work on object detection task.

---

> ### Author Response · Authors · 2017-12-08
> **Comments**
>
> We acknowledge the reviewer’s comments regarding experiments on larger datasets and more accurate baseline models. We chose to run on CIFAR-10 and OpenNMT since they have reasonable iteration times. This allows us to carefully study the limits of the methods, and the trade-offs between bit width, network depth, and network layer width, given a fixed model by performing several experiments at each data point in reasonable time. To address the reviewer’s point, we are extending our experiments to training
> 1) quantized ResNet students, e.g. ResNet18, from larger ResNet teachers, e.g. ResNet50, on ImageNet, comparing against the full-precision and existing quantized state-of-the-art baselines.
> 2) quantized state-of-the-art students for CIFAR-10 and CIFAR-100 tasks from the full-precision baselines, and comparing them against best performing published quantized versions.
>
> It would be very helpful if the reviewer could be more precise with respect to what they consider as a good baseline.
>
> Regarding the performance of differentiable quantization (DQ) with respect to quantized distillation (QD), we point out that
> 1) in constrained settings (e.g. 2bit quantization) DQ can be significantly more accurate than QD. See for example the CIFAR-100 2-bit experiment. We will add more examples here.
> 2) DQ usually converges earlier than QD to similar accuracy, which can be useful in settings where reducing number of iterations is important.

---

> ### Author Response · Authors · 2017-12-30
> **New revision**
>
> Thank you for your patience. Please see the reply above and the new revision of the paper.

---

### Official Review · AnonReviewer2 · 2017-11-27
**Review of "Model compression via distillation and quantization"**

**Rating:** 7
**Confidence:** 2

**Review:**

The paper proposes to combine two approaches to compress deep neural networks - distillation and quantization. The authors proposed two methods, one largely relying on the distillation loss idea then followed by a quantization step, and another one that also learns the location of the quantization points. Somewhat surprisingly, nobody has combined the two approaches before, which makes this paper interesting. Experiments show that both methods work well in compressing large deep neural network models for applications where resources are limited, like on mobile devices.

Overall I am mostly OK with this paper but not impressed by it.  Detailed comments below.

1. Quantizing with respect to the distillation loss seems to do better than with the normal loss - this needs more discussion.
2. The idea of using the gradient with respect to the quantization points to learn them is interesting but not entirely new (see, e.g., "Matrix Recovery from Quantized and Corrupted Measurements", ICASSP 2014 and "OrdRec: An Ordinal Model for Predicting Personalized Item Rating Distributions", RecSys 2011, although in a different context). I also wonder if it would work better if you can also allow the weights to move a little bit (it seems to me from Algorithm 2 that you only update the quantization points). How about learning them altogether? Also this differentiable quantization method does not really depend on distillation, which is kind of confusing given the title.
3. I am a little bit confused by how the bits are redistributed in the second method, as in the end it seems to use more than the proposed number of bits shown in the table (as recognized in section 4.2). This makes the comparison a little bit unfair (especially for the CIFAR 100 case, where the "2 bits" differentiable quantization is actually using 3.23 bits). This needs more clarification.
4. The writing can be improved. For example, the concepts of "teacher" and "student" is not clear at all in the abstract - consider putting the first sentence of Section 3 in there instead. Also, the first sentence of the paper reads as "... have showed tremendous performance", which is not proper English. At the top of page 3 I found "we will restrict our attention to uniform and non-uniform quantization". What are you not restricting to, then?

Slightly increased my rating after reading the rebuttal and the revision.

---

> ### Author Response · Authors · 2017-12-08
> **Comments**
>
> 1. The importance of distillation loss:
> Across all our experiments, using distillation loss in quantizing was superior to using ‘normal’ loss. This is one of our main findings, and is extremely consistent across experiments. We have given two sets of experiments to illustrate this, but we will add more examples.
> 2. Differentiation w.r.t. quantization points:
> We will discuss the connection with the RecSys 2011 and ICASSP papers in the next revision.
> Alternating optimization w.r.t. weights and locations of quantization points is a neat idea, which we are experimenting with; we will present results on it in the next revision.
> Distillation is actually present in Differentiable Quantization (DQ), since we are starting from a distilled full-precision model. It is true however that DQ can be applied independently from distillation. We will clarify this point.
> 3. Re-distribution of bits:
> Fractional numbers appear since there are a couple of different techniques being concurrently: 1) we preferentially re-distribute bits proportionally to the gradient norm; 2) we perform Huffman encoding to compute the “optimal” resulting compression score per layer; 3) we do bucketing, which slightly increases the bit cost due to storing the scaling factor.
> We will add a detailed procedure on how these costs were obtained, and explain why they can exceed the baseline bit width.
> 4. Writing inconsistencies:
> We thank the reviewer for the detailed comments, which we will fully address in the next version.

---

> ### Author Response · Authors · 2017-12-30
> **New revision**
>
> Thank you for your patience. Please see the reply above and the new revision of the paper.

---

### Official Review · AnonReviewer3 · 2017-11-29
**A very interesting paper with good motivation and good results.**

**Rating:** 8
**Confidence:** 5

**Review:**

This paper presents a framework of using the teacher model to help the compression for the deep learning model in the context of model compression. It proposed both the quantized distillation and also the differentiable quantization. The quantized distillation method just simply adapt the distillation work for the task of model compression, and give good results to the baseline method. While the differentiable quantization optimise the quantization function in a unified back-propagation framework. It is interesting to see the performance improvements by using the one-step optimisation method.

I like this paper very much as it is in good motivation to utilize the distillation framework for the task of model compression. The starting point is quite interesting and reasonable. The information from the teacher network is useful for constructing a better compressed model. I believe this idea is quite similar to the idea of Learning using Privileged Information, in which the information on teacher model is only used during training, but is not utilised during testing.

Some minor comments:
In table 3, it seems that the results for 2 bits are not stable, and are there any explanations?
What will be the results if the student model performs the same with the teacher model (e.g., use the teacher model as the student model to do the compression) or even better (reverse the settings)?
What will be the prediction speed for each of models? We can also get the time of speedup for the compressed model.

It will be better if the authors could discuss the connections between distillation and the recent work for the Learning using Privileged Information setting:
Vladimir Vapnik, Rauf Izmailov:
Learning using privileged information: similarity control and knowledge transfer. Journal of Machine Learning Research 16: 2023-2049 (2015)
Xinxing Xu, Joey Tianyi Zhou, IvorW. Tsang, Zheng Qin, Rick Siow Mong Goh, Yong Liu: Simple and Efficient Learning using Privileged Information. BeyondLabeler: Human is More Than a Labeler, Workshop of the 25th International Joint Conference on Artificial Intelligence (IJCAI-16). New York City, USA. July, 2016.

---

> ### Author Response · Authors · 2017-12-08
> **Comments**
>
> 1. Divergence of 2bit variants:
> Indeed, the 2bit version can diverge for some parameter settings. Our interpretation is that through trimming and quantization we reduce the capacity of the student, which might no longer have enough capacity to mimic the teacher model, and diverges.
> 2. Swapping student and teacher model:
> That is an interesting question. We focused on larger teachers and smaller students for compressing a fixed model. However, it is highly probable that quantizing a larger CNN via distillation from a smaller one will work as well. We will add experiments for this case.
> 3. Inference speed is linear in network depth and bit width. In our experiments, the speedup we get on inference is proportional to the reduction in depth. This is mentioned in passing in the Conclusions section, but we will add some exact numbers.
> 4. Connection to “Learning using Privileged Information”:
> This is an excellent point, we will discuss this connection in the next revision.

---

> ### Author Response · Authors · 2017-12-30
> **New revision**
>
> Thank you for your patience. Please see the reply above and the new revision of the paper.

---

### Public Comment · ~bruce_matthew_kuzak1 · 2017-11-29
**Source Code**

I was wondering if you guys have an open source code for your experiment along with a the data used for training and validating that we could use to reproduce your results.

Me and my team, would like to revise your research paper for a final class project.

thank you

---

> ### Author Response · Authors · 2017-12-05
> **Sure**
>
> Sorry for the late reply, we had to check what would be the best way to share the code without breaking anonymity. If you tell me your GitHub username, I can add you to the repository :)

---

> > ### Public Comment · ~Raden_Muaz1 · 2018-01-02
> > **Not OP, but can I look into source code?**
> >
> > I am interested into reproducing your results, and applying it for LSTM and GRU
> > my github is https://github.com/RadZaeem

---

> > > ### Author Response · Authors · 2018-01-02
> > > **Invitation sent**
> > >
> > > Hi,
> > >
> > > sure! I have added you to the github repo :) Note that in the paper we report experiments with LSTM for German-English translation, on the openNMT integration test dataset and WMT13.

---

> > ### Public Comment · ~Hui_Xiang1 · 2018-04-12
> > **source code**
> >
> > Hey, I am also very interested on this paper, can you share me your source code? my github is  https://github.com/PangHua, thanks very much.

---

> > > ### Author Response · Authors · 2018-04-12
> > > **Code has been published**
> > >
> > > Hi,
> > >
> > > Thank you for your interest! The code has already been published on github. The link is reported in the latest revision of the paper; anyhow you can find it at https://github.com/antspy/quantized_distillation

---

> > > > ### Public Comment · ~Hui_Xiang1 · 2018-04-13
> > > > **Thanks.**
> > > >
> > > > Ur... Sorry, I just saw the title and abstraction is very attractive, haven't able to looked it over.

---

### Author Response · Authors · 2017-12-08
**Thank you to reviewers**

We would like to thank all the reviewers for their careful consideration of our paper, and for very useful comments. We provide detailed responses below.
We are currently running additional experiments to address some of the reviewers’ comments. We plan to produce a complete updated revision as soon as the experiments are done, which should be before December 15th.

---

### Author Response · Authors · 2017-12-30
**New revision**


We have posted a revision covering the reviewers comments. We detail the main changes below:

- Larger-scale experiments:

To address the comments by Reviewers 1 and 3, we have added experiments on the ImageNet-1K and WMT13 datasets, as well as extended the CIFAR experiments.

On ImageNet, we trained ResNet-18 at 4bits of precision, distilling from a ResNet-34 teacher.
Direct quantized disillation produces a 4-bit ResNet18 model which loses around 4% in terms of top-1 accuracy w.r.t. the 32-bit ResNet-18 baseline, which is in line with the performance of existing techniques. To improve upon this, we investigated widening the convolutional layers of the student (a technique known to increase capacity during distillation). This leads the student to match and surpass the accuracy of the baseline.
In the revised paper, we detail a new experiment where we obtain a 4-bit quantized ResNet-18 student matching the accuracy of the ResNet-34 teacher, and improving upon the accuracy of the 32-bit ResNet-18 of the same depth by >3%. The resulting model is < 1/4th the size of ResNet-34, has virtually the same accuracy, and, thanks to decreased depth, it is >50% faster in terms of inference speed.
Similarly, on WMT13, we present a 4bit model which matches the accuracy of the 32-bit teacher in terms of both BLEU score and perplexity.
In addition, on CIFAR-10 and CIFAR-100, we derive quantized variants of state-of-the-art models (wide ResNets) which show almost no loss in accuracy. See section 6 for details.

In sum, we believe that these results validate our techniques in settings that are close to the state-of-the-art, and hope that they address the reviewers’ questions.


- Related work:

We have significantly expanded the related work section, to clarify the relation to the papers which the reviewers suggested, and to the “Binarized Neural Networks on the ImageNet Classification Task” paper. In particular, we have clarified the fact that distillation for size reduction has been suggested before (even since the paper by Hinton et al.). Our contribution is in significantly refining these techniques to (almost) eliminate all accuracy loss in the context of image classification and machine translation using DNNs. Please see the related work section in the introduction for details.

- Clarifications in the presentation:

We have performed an in-depth revision to clarify the reviewers’ various questions, along the lines of our initial rebuttal. The reviewers can view individual changes via the PDF diff. We have also added a few smaller experiments to clarify various reviewer questions, such as training a student model deeper than the teacher (see Section 6). We also tried alternating QD with DQ, but experiments suggested this method to produce inferior results, so we did not investigate it further.

Finally, we would like to apologize for the delayed revision. This was due to technical issues in the default training setup for ResNet on the training framework we employed, as well as to the fact that ImageNet experiments took longer than expected.

---

> ### Author Response · Authors · 2018-01-05
> **Second Revision**
>
> We have performed a second revision of the above, further refining the presentation, and including an additional ImageNet experiment, in which we distil onto a wide 4-bit ResNet34 model from ResNet50.
> The results confirm our earlier findings for ResNet18: we are able to recover the accuracy of the teacher (50 layers, full precision, 76.13% top-1 accuracy) into a shallower quantized student (34 layers, 4-bit weights, 76.07% top-1 accuracy), using quantized distillation. The results are state-of-the-art for this combination of parameters, and surpass the accuracy of the standard full-precision ResNet34 (73.6% top-1 accuracy).

---

### Decision · Program_Chairs · 2018-01-29
**ICLR 2018 Conference Acceptance Decision**

**Decision:**

Accept (Poster)

**Comment:**

The submission proposes a method for quantization.  The approach is reasonably straightforward, and is summarized in Algorithm 1.  It is the analysis which is more interesting, showing the relationship between quantization and adding Gaussian noise (Appendix B) - motivating quantization as regularization.

The submission has a reasonable mix of empirical and theoretical results, motivating a simple-to-implement algorithm.  All three reviewers recommended acceptance.